# Multi-User MultiWOZ: Task-Oriented Dialogues among Multiple Users

**Yohan Jo**[1]      **Xinyan Zhao**[2]      **Arijit Biswas**[2]      **Nikoletta Basiou**[2]      **Vincent Auvray**[2]
**Nikolaos Malandrakis**[2]      **Angeliki Metallinou**[2]      **Alexandros Potamianos**[2]

[1]Seoul National University, Korea      [2]Amazon, USA

yohan.jo@snu.ac.kr

{xinyazha,barijit,nbasiou,vauvray,malandrn,ametalli,potamian}@amazon.com

## Abstract

While most task-oriented dialogues assume conversations between the agent and one user at a time, dialogue systems are increasingly expected to communicate with multiple users simultaneously who make decisions collaboratively. To facilitate development of such systems, we release the **Multi-User MultiWOZ** dataset: task-oriented dialogues among two users and one agent. To collect this dataset, each user utterance from MultiWOZ 2.2 was replaced with a small chat between two users that is semantically and pragmatically consistent with the original user utterance, thus resulting in the same dialogue state and system response. These dialogues reflect interesting dynamics of collaborative decision-making in task-oriented scenarios, e.g., social chatter and deliberation. Supported by this data, we propose the novel task of **multi-user contextual query rewriting**: to rewrite a task-oriented chat between two users as a concise task-oriented query that retains only task-relevant information and that is directly consumable by the dialogue system. We demonstrate that in multi-user dialogues, using predicted rewrites substantially improves dialogue state tracking without modifying existing dialogue systems that are trained for single-user dialogues. Further, this method surpasses training a medium-sized model directly on multi-user dialogues and generalizes to unseen domains.[1]

## 1  Introduction

Voice assistants like Amazon Alexa and Google Assistant are widespread, and users often interact with them in multiparty settings, such as playing games and making decisions with family members (Porcheron et al., 2018). However, most dialogue systems are designed to support only single-user dialogues, i.e., the agent expects to converse with one user at a time via a succinct command that contains

Figure 1: Excerpts of dialogues in our dataset. For each example, the sequence of user utterances is called **multi-user chat**. Rewrites refer to original user utterances in MultiWOZ 2.2 that were expanded to multi-user chats between User1 and User2. They can be used as ground-truth rewrites for contextual query rewriting. More examples are in Appendix A.1.

all and only necessary information for conducting a task. By contrast, multi-user task-oriented dialogues are significantly richer, containing deliberation and social chatter, and poses additional challenges like separating out task-relevant information from social and sensitive information related to user privacy. The main bottleneck for a first step into supporting multi-user task-oriented dialogues is a lack of proper datasets, as most (if not all) existing datasets for task-oriented dialogues are single-user. To overcome this limitation and facilitate future research, we build and release a dataset of multi-user task-oriented dialogues and propose the novel task

---

[1]The dataset is available at https://github.com/yohanjo/multiuser_multiwoz.

of multi-user contextual query rewriting.

Our dataset **Multi-User MultiWOZ** is an extension of MultiWOZ 2.2 (Zang et al., 2020) to multi-user dialogues (Figure 1, §3). MultiWOZ 2.2 is one of the largest and most popular single-user task-oriented dialogues. The guiding principle of our data collection is to extend each user utterance in MultiWOZ 2.2 to a chat between two users making decisions together (**multi-user chat** henceforth) that leads to the same dialogue state as the source utterance. This allows us to (1) re-use system acts and responses annotated in the source single-user dialogue and (2) train a query rewriting model that converts a multi-user chat to its source utterance (as the ground-truth rewrite) so that output rewrites can be consumed by dialogue systems that expect single-user utterances. Compared to existing related datasets, which are either single-user (Andreas et al., 2020; Rastogi et al., 2020; Young et al., 2021) or lack dialogue state annotations (Li et al., 2017), our dialogues reflect interesting dynamics of collaborative decision-making in task-oriented conversations, such as questions to elicit slot values, social chatter, and deliberation.

Empowered by this dataset, we propose the novel task of **multi-user contextual query rewriting**: to rewrite a multi-user chat as a single request that is concise and contains all and only task-relevant information (§5). This task is important because (1) it can bridge the gap between multi-user chats and a dialogue system trained for single-user dialogues without replacing the entire dialogue system, and (2) it alleviates users' privacy concerns by processing multi-user chats on device and sending the rewrites to the system server with only task-relevant information. We demonstrate the accuracy of baseline models on the rewriting task and discuss the main challenges. Further, we verify that model-predicted rewrites are helpful for dialogue state tracking for dialogue systems that are trained on single-user dialogues, by substantially outperforming a baseline that simply concatenates utterances in a multi-user chat and treats it as a "single" utterance. This task also benefits unseen domains.

Our contributions are twofold:

- We release the **Multi-User MultiWOZ** dataset, task-oriented dialogues between two users and one agent, under the MIT license.
- We propose the **multi-user contextual query rewriting** task that rewrites multi-user chats as concise task-oriented requests.

## 2 Related Work

Our work is closely related to three fields of NLP: task-oriented dialogues, contextual query rewriting, and dialogue summarization.

**Task-Oriented Dialogues:** Existing datasets of task-oriented dialogues (Zang et al., 2020; Andreas et al., 2020; Byrne et al., 2019; Rastogi et al., 2020; Zhu et al., 2020; Young et al., 2021) are, to our knowledge, all single-user conversations. By contrast, our dataset consists of task-oriented dialogues where two users are making decisions together with the help of an agent. Since input to a dialogue system is a chat between users rather than a single user utterance, such dialogues pose challenges to dialogue systems trained via traditional algorithms for dialogue state tracking (Hosseini-Asl et al., 2020; Le et al., 2020a,b). While multiparty dialogue datasets exist (Li et al., 2017; Gopalakrishnan et al., 2019), they are not task-oriented and thus missing important information for training task-oriented dialogue systems. Our multi-user dataset is task-oriented by nature and annotated with relevant information, such as dialogue states and system acts and responses.

**Contextual Query Rewriting:** Contextual query rewriting refers to decontextualizing a user query to a self-contained one that contains necessary contextual information for the agent to serve the user's request (Zamani et al., 2022). This mostly involves anaphora resolution and ellipsis resolution, and sequence-to-sequence models (e.g., GPT) have performed well (Vakulenko et al., 2021; Yu et al., 2020). Our work is related in that the source utterance of each mutli-user chat can be seen as a decontextualized rewrite of the chat that the agent can process without parsing the whole chat. While existing work and datasets for contextual query rewriting focus on converting one user request at a time (Yuan et al., 2022; Dalton et al., 2009; Choi et al., 2018), our task rewrites a chat between two users (hence, *multi-user* contextual query rewriting). This task is more challenging as it often goes beyond anaphora/ellipsis resolution; for instance, it also involves resolving deliberation between users, and user intents and slots may be spread out across multiple utterances in a chat. Our dataset provides training and evaluation resources for this task.

**Dialogue Summarization:** Our dataset is also related to dialogue summarization in that each multi-

user chat can be seen as a short task-oriented dialogue between two users and its rewrite as a summary of the dialogue. Existing datasets for dialogue summarization are not well-suited for summarizing multi-user task-oriented dialogues. One of the main reasons is that they have different goals for summarization. For instance, many datasets (Gliwa et al., 2019; Krishna et al., 2021; Zhang et al., 2021; Song et al., 2020; Zhong et al., 2021; Chen et al., 2021; Zhu et al., 2021) aim to summarize diverse perspectives of speakers rather than succinct task-oriented information (Fabbri et al., 2021; Lin et al., 2022). Our dataset focuses on summaries where deliberations are resolved and only task-relevant information (e.g., user intents and slots) is retained. While some datasets aim to summarize task-oriented dialogues from customer services (Zhao et al., 2021; Liu et al., 2019; Feigenblat et al., 2021; Lin et al., 2021), they are not designed to summarize chats between users; rather, they summarize a chat between user and agent, which has different dynamics than collaborative decision-making between users.

## 3 Data Collection

In building a dataset of multi-user task-oriented dialogues, our principle is to extend an existing dataset of single-user task-oriented dialogues, such that each user utterance in that dataset is expanded to a chat between two users (**multi-user chat**) that leads to the same dialogue state. Two main benefits to this approach are: (1) we can reuse system acts and responses in the original dataset without annotating them anew, and (2) pairs of a multi-user chat (in the collected data) and its source utterance (in the original data) can be used to train a query rewriting model that rewrites a multi-user chat as a concise query (and vice versa).

To that end, we use MultiWOZ 2.2 (Zang et al., 2020) as our basis[2]. It consists of task-oriented dialogues between one user and one agent, where the agent helps the user with booking and finding information for eight services (hotel, attraction, restaurant, train, taxi, bus, police, hospital). We expand each of the first four user utterances[3] in each dialogue to an imaginary chat between two users making collaborative decisions.

We describe our pilot study, data collection protocol, validation process, and data statistics.

---

[2]MultiWOZ 2.2 is distributed under the MIT license.

[3]We focus only on four user utterances per dialogue because of a limited budget. This decision does not impair the representation of services, intents, or slots (Appendix A.2).

### 3.1 Pilot Study

We first ran a pilot study to learn the characteristics of collaborative decision-making in task-oriented dialogues and make sure they are reflected in our generated chats. Specifically, we recruited Alexa users and asked them to conduct two tasks in pairs using two Alexa skills. The first is GiftFinder where the users navigate gifts to buy, and the second is NewsFinder where the users find news articles of common interest. The pilot data revealed four notable characteristics of multi-user chats. Users (1) ask questions to each other to elicit slot values (e.g., "What time are we thinking?"), (2) have social chatter aside from expressing intents and slot values (e.g., "How are you going to fix my empty stomach?"), (3) have deliberation over multiple options (e.g., "No, too many stops. Let's take a taxi."), and (4) exploit common ground, e.g., mention the names of each other and friends (e.g., "Steve can't get there till after 19:00.").

### 3.2 Data Collection Protocol

To collect multi-user dialogues at scale, we use Amazon Mechanical Turk (MTurk). Each task consists of the first four user utterances of a dialogue from MultiWOZ (Figure 6–8 in Appendix B.2). Each utterance is accompanied by the system response. The number of utterances in the generated multi-user chat is predefined between 2 and 4, being skewed toward the number of informed or requested slots (Appendix B.3). We asked one turker to expand all user utterances in each task. Compared to having two turkers do that together, this is faster and still results in high quality, as will be discussed in our dialogue quality assessment (§4). Tasking one worker to generate the entire dialogue has been shown in prior research to be a simple approach that leads to better dialogue quality (Young et al., 2021; Byrne et al., 2019; Nakamura et al., 2022).

To ensure that a generated chat preserves the dialogue state of the original (source) utterance, we constrained that all informed slot values and requested slots be mentioned in the generated chat (see Appendix B.4). Note that this makes the generated chats compatible with the system acts and responses in MultiWOZ.

For generated chats to reflect the characteristics revealed in the pilot study, our instructions included example dialogues that contain social chatter and deliberations. Turkers naturally elicited slot values

and mentioned names.

For the dev and test sets in MultiWOZ, we covered all dialogues (1,000 each). For the training set, we sampled 2,400 dialogues while including all dialogues about the bus, hospital, and police services (since these services are highly underrepresented). Turkers did not overlap between the training set and the test set.

Table 1 shows some example dialogues. More details about the data collection task are available in Appendix B.

### 3.3 Validation

We validated generated multi-user chats via a separate MTurk task. Each task contains one dialogue, i.e., at most four multi-user chats. Each chat is accompanied by the previous system response (except for the first turn) and the source utterance labeled as "summary" (Figure 13–15 in Appendix C.2).

For each pair of generated chat and summary, we asked the following questions.

1. Is the users' relationship two customers making decisions together? Yes or no?
2. Is the chat a realistic chat between humans? Very much, acceptable, or definitely not?
3. Is the last utterance in the chat a realistic utterance toward the agent? Very much, acceptable, or definitely not?
4. Does the summary have any missing information that is present in the chat? Yes or no? If yes, what information is missing?
5. Does the summary contain any information that is not present in the chat? Yes or no? If yes, what additional information is present?

We curated 13 qualified validators through a qualification task that ensured the validators could apply the five criteria correctly (Figure 9–12).

Next, every chat-summary pair in our collection was validated by two qualified validators, where it is considered "flagged" if a validator chooses "no" or "definitely not" for Q1–Q3 or "yes" for Q4–Q5. Based on the results, each chat-summary pair is categorized into: **poor** if flagged by both validators for at least one question, **good** if not flagged by any validators for any questions, and **moderate** otherwise. Further, a dialogue is categorized as **poor** if any of its chats is poor, **good** if all its chats are good, and **moderate** otherwise. Poor dialogues are discarded from the final dataset. More details about the tasks are available in Appendix C.

Based on the validation results, we computed the score for each criterion as follows. The questions have the following option-score mapping: Yes=1, No=0 / Very-much=2, Acceptable=1, Definitely-Not=0. Overall, the final scores are satisfactory: (Q1) 0.99/1 (Q2–3) 1.8/2, (Q4–5) 0.98/1. Especially the scores of Q4–5 suggest that dialogue states are preserved between multi-user chats and their source utterances.

The five validation criteria we used are important quality metrics for multi-user task-oriented dialogues. We see an opportunity to automate some of them; for example, to automatically check whether dialogue participants have a relationship of two customers, we could train a classifier that distinguishes among different types of participant relationships using existing dialogue datasets. We leave this to future work.

### 3.4 Data Statistics

Table 2 shows some statistics of our data. Compared to the single-user counterparts in MultiWOZ, our dataset contains 2.7x turns, 1.7x tokens, and 1.3x negations. Based on a random sample of 300 multi-user chats (100 for each split), we counted the occurrences of three important types of social dynamics that we found to be characteristic of multi-user task-oriented dialogues:

- Slot elicitation: Users ask each other for information or preferences related to intents and slots (e.g., "Let me think, there are how many in our group?").
- Social chatter: Utterances for social conversation, such as jokes, feelings, and more (e.g., "I'm not sure, but I've heard good things."). 'Social chatter' is a broad concept that could be further broken down into intents or dialogue acts specific to multiparty dialogues (e.g., suggesting why a certain slot value is preferable as in "too many stops" in Figure 1). Such a breakdown is beyond the scope of this paper.
- Deliberation: Users make a decision over multiple options (e.g., User1: "Either 4–5 not sure." → User2: "Better make it five to be safe.").

More examples are available in (Appendix A.3 and Figure 1). We found that slot elicitation appears in 24%, social chatter in 23%, and deliberation in 2% of multi-user chats.

Our dataset has a reasonable size, containing 16,706 multi-user chats, which is slightly larger

| Dialogue 1 | Dialogue 2 |
|---|---|
| **User2**: Should be try the Alpha-Milton Guest House? | **User1**: We need your help in finding a local hospital. |
| **User1**: What do we know about it? | **User2**: Can you find one with a cardiology department? |
| **User2**: Not much. Can you give us some info about the place? | **RW**: Is there a local hospital that has a cardiology department? |
| **RW**: Can you please tell me about the Alpha-Milton Guest House? | **Sys.**: Addenbrookes Hospital located at Hills Rd, Cambridge has cardiology. Do you need their phone number? |
| **Sys.**: Absolutely! They are a wonderful guesthouse in the northern part of town. They are moderately priced and rated 3 stars. | **User1**: Let's not waste time. |
| **User2**: We do not want to pay for parking. | **User2**: I'm not. But it's important to get the relevant information. |
| **User1**: Does it have free parking? | **User1**: Sorry, I didn't mean that. But we need to hurry. |
| **RW**: Do they have free parking? | **User2**: I know. Can you just give us their address and postcode? |
| **Sys.**: No they unfortunately do not have parking. Is there anything else i can help you with or any other information you would like to know about that hotel or area? | **RW**: I just need their address and postcode please |
| **User1**: No, that is fine. Should we take a train or bus to cambridge? | **Sys.**: Their address is Hills Rd, Cambridge, postcode CB20QQ |
| **User2**: Train would better. | **User1**: Thanks a lot. |
| **User1**: Alright. Please get us a train for Tuesday. | **User2**: You've been very helpful. |
| **RW**: No, thats ok. I do need a train to Cambridge on Tuesday though. | **User1**: We gotta go now. |
| **Sys.**: Where are you departing from? And what time would you like to arrive? | **User2**: Goodbye. Have a great day. |
| **User2**: We should leave from london kings cross in the afternoon. | **RW**: Thank you and goodbye. |
| **User1**: Yes, we need to get there by 3pm. | **Sys.**: You're welcome. I am glad I could assist you at TownInfo centre. Goodbye. |
| **User2**: Alright. Please book a train that arrives by 15:30. | |
| **RW**: I need to arrive by 15:30 and am leaving london kings cross. | |
| **Sys.**: TR3456 leaves London Kings Cross on Tuesday at 13:17 and arrives in Cambridge at 14:08. The cost is 23.60 pounds. Will this meet your needs? | |

Table 1: Example dialogues. (**RW**: rewrite (i.e., source utterance), **Sys.**: system response)

than a popular dialogue summarization dataset SAMSum (16,369) (Gliwa et al., 2019) and a parallel corpus for contextual query rewriting TREC CAsT (173) (Dalton et al., 2009).

Note that each multi-user chat has the labels of informed slot-value pairs, requested slots, and intents, because the original dialogue state is preserved through text matching of informed/requested slots (§3.2) and semantic validation to contain no missing or extra information (§3.3). This allows us to identify the text spans of slots and values via text match.

Multi-user chats in our dataset contain 2 to 4 utterances, which does not handle cases where a user speaks to the agent without discussing with the other user. Our data collection did not cover such cases because user utterances in the original MultiWOZ are already reflecting such scenarios. Hence, we recommend training a dialogue system on a mix of our dataset and the original MultiWOZ so that the system can process both single user utterances and multi-user chats reliably.

## 4 Dialogue Quality Assessment

We verify that the collected dialogues have high quality and are realistic to happen in the real world. Adapting the dialogue quality assessment in Chen et al. (2023) to multi-user dialogues, we evaluated each dialogue in six aspects:

1. **Realistic:** How likely would the user chats occur in real-world interactions with an agent? Scale: 1 (completely unlikely to occur) to 5 (highly likely to occur)
2. **Natural:** How fluent are the user chats? Scale: 1 (completely influent) to 5 (as fluent as native English speakers)
3. **Coherent:** How coherent is the overall flow of the dialogue? Scale: 1 (completely incoherent) to 5 (as coherent as reasonably intelligent and attentive speakers)
4. **Interesting:** How interesting are the user chats? Scale: 1 (generic and dull) to 5 (full of content and very engaging)
5. **Consistent:** How consistent is each user? Scale: 1 (always says something that abruptly

| | Train | | Dev | | Test | |
|---|---|---|---|---|---|---|
| | MultiWOZ | MultiUserWOZ | MultiWOZ | MultiUserWOZ | MultiWOZ | MultiUserWOZ |
| # of dialogues (good/moderate) | – | 1,589/695 | – | 793/202 | – | 731/263 |
| # multi-user chats | – | 8,859 | – | 3,936 | – | 3,911 |
| # of Turns/Dialogue | 4 | 11 (2.8x) | 4 | 11 (2.7x) | 4 | 11 (2.7x) |
| # of Tokens/Dialogue | 55 | 104 (1.9x) | 56 | 103 (1.8x) | 56 | 92 (1.7x) |
| # of Tokens/Turn | 14 | 10 (0.7x) | 14 | 10 (0.7x) | 14 | 9 (0.6x) |
| # of Stopwords | 63,128 | 119,954 (1.9x) | 28,117 | 52,833 (1.9x) | 27,869 | 47,213 (1.7x) |
| # of Entities | 6,614 | 8,771 (1.3x) | 3,089 | 4,237 (1.4x) | 3,133 | 3,716 (1.2x) |
| # of Negations | 260 | 317 (1.2x) | 102 | 194 (1.9x) | 88 | 112 (1.3x) |

Table 2: Data statistics. "MultiWOZ" columns shows the statistics of single-user counterparts in MultiWOZ.

| | Real | Nat | Coh | Int | Cons | Rel |
|---|---|---|---|---|---|---|
| Multi²WOZ | 4.47 | 4.67 | 4.48 | 3.79 | 4.46 | 4.59 |
| MultiWOZ | 4.68 | 4.83 | 4.68 | 3.72 | 4.69 | 4.71 |
| Pilot | 4.20 | 4.57 | 4.43 | 3.55 | 4.46 | 4.41 |
| DailyDialog* | – | 4.85 | 4.51 | 3.44 | 4.57 | – |
| TopicalChat* | – | 4.92 | 4.39 | 4.55 | 4.87 | – |

Table 3: Dialogue quality assessment. **Multi²WOZ** refers to our dataset Multi-User MultiWOZ. (**Real**istic, **Nat**ural, **Coh**erent, **Int**eresting, **Cons**istent, **Rel**event) *The scores of these datasets are from Chen et al. (2023).

contradicts what they said earlier) to 5 (never says something that abruptly contradicts what they said earlier)

6. **Relevant:** Are the user chats relevant to the given scenario? Scale: 1 (completely irrelevant) to 5 (highly relevant)

We randomly sampled 90 dialogues from our data for assessment. For baselines, we also evaluated two reference datasets for comparison: (1) the same 90 dialogues from original MultiWOZ 2.2 (single-user). In this case, the quality of user utterances (as opposed to user chats) is assessed; and (2) 62 pairwise dialogues from our pilot study (§3.1) that were generated by participants in pairs. Each dialogue was evaluated by three qualified turkers.

Table 3 lists the quality scores of the datasets. The table also reports the scores of two well-established multiparty dialogue datasets: DailyDialog (Li et al., 2017) and TopicalChat (Gopalakrishnan et al., 2019), evaluated by Chen et al. (2023).

For Realism, our data and original MultiWOZ data showed no statistically significant difference (Mann-Whitney U test), meaning evaluators judge our dialogues to be as likely to occur in reality as the well-established MultiWOZ. Our data was rated higher than the pilot data (p-value=0.03), suggesting that collecting multi-user task-oriented dialogues in a pairwise setting is difficult. It can produce less realistic dialogues than our protocol, mainly because laypeople are not good at creating a collaborative scenario and holding a relevant dialogue in interactive settings, unlike single-user settings like MultiWOZ and Wizard Of Wikipedia (Dinan et al., 2019).

Regarding the other criteria, original MultiWOZ was rated slightly higher than our data (by 0.2 points) for Naturalness, Coherence, and Consistency with p-value < 0.05. This is expected because it is naturally difficult for multi-user dialogues to achieve the same level of Coherence and Consistency as single-user dialogues. Compared to the pilot data, our data showed no statistically significant difference for any criteria (except for Realism), suggesting our protocol produces similar quality to a pairwise setting. Even compared to well-established multi-party dialogue datasets, our dialogues are scored higher than DailyDialog for Interestingness and TopicalChat for Coherence.

# 5 Multi-User Contextual Query Rewriting

We propose the novel task of multi-user contextual query rewriting, which rewrites a task-oriented chat between users as a concise query. Our main goal in creating this task is for this query rewriting to serve as the first module of a language understanding pipeline. The main advantage of such a module is that it can be plugged into an existing dialogue system and convert a multi-user chat to a concise self-contained utterance, enabling multi-user dialogues with minimal modifications to the dialogue system. Further, this module can filter out task-irrelevant information (e.g., personal information and social chatter) on device before sending the rewrites to the system server, thus enhancing user privacy.

|            | R-1  | R-2  | R-L  | Inf  | Req  | Hal  |
|------------|------|------|------|------|------|------|
| T5-Base    | 53.8 | 33.1 | 49.3 | 94.7 | 92.4 | 18.9 |
| BART-Base  | **56.0** | **35.8** | **52.1** | 93.0 | **92.9** | 16.7 |
| BART-Large | **56.0** | 35.3 | 51.6 | **95.4** | 92.5 | **16.5** |
| GPT-2-Base | 42.2 | 23.1 | 38.5 | 75.6 | 67.8 | 28.5 |

Table 4: Query rewriting accuracy. (**R**OUGE, **Inf**ormed slot values, **Req**uested slot names, **Hal**lucination Rate)

## 5.1 Experiment Settings

We explore four baseline models: T5-base (Raffel et al., 2019), GPT-2-base (Radford et al., 2019), BART-base and BART-large (Lewis et al., 2019), via the HuggingFace library (Wolf et al., 2020). We chose moderate-sized models (the '-base' models) as our main baselines, considering limited hardware resources in popular smart speakers (e.g., Amazon Echo Dot). We included BART-large to see if a larger model improves accuracy. Hyperparameters are detailed in Appendix D.1.

The input is a concatenation of all utterances in a multi-user chat, each of which is prefixed with "<user>", e.g., "<user>Shall we take a bus from Saint John's college to Pizza Hut Fen Ditton?<user>No, too many stops. Let's take a taxi.<user>Can you book a taxi for us?". We do not add dialogue history because it was not helpful. The output is a rewrite of the chat; the source utterance of the chat is used as the ground truth. We use all dialogues labeled 'good' or 'moderate' for the train and dev data. Results are reported on test dialogues labeled 'good'.

We report ROUGE-1, 2, and L scores (Lin, 2004)–popular summarization accuracy metrics. Since the coverage of informed and requested slots is key to this task, we also report the recall of informed slot values and requested slot names. Lastly, we measure hallucination rates as the inverse of entity precision between true and predicted rewrites, where entities are extracted using spaCy[4].

## 5.2 Query Rewriting Results

Table 4 shows the query rewriting accuracy of the models. Overall, BART models perform best across most metrics, followed by T5. GPT-2 performs significantly worse as it tends to make up stories that are absent in the input chats; this results in low recall of informed and requested slots and high hallucination rates. BART-large are on par with BART-base; while it has a higher recall of In-

---

[4] https://spacy.io/models/en#en_core_web_trf

---

| User1: Ok, and the postcode? You haven't told us the postcode yet. |
| --- |
| **User2:** Seems like the fun time is secured. But how are you going to fix my empty stomach? |
| **User1:** I know you love indian food, and I feel like eating some, too. |
| **User2:** So check for an indian restaurant that is not far from our nightclub of choice. |
| **Ground-Truth Rewrite (Source Utterance):** What is the postcode for that? I am also looking for an indian restaurant near the nightclub, are there any? |
| |
| **T5:** i'm looking for an indian restaurant that is not far from my nightclub of choice. |
| **BART:** what is the postcode? |
| **GPT2:** can you help me find an indian restaurant located in a college or a hotel room that i am in? |

Table 5: Example outputs of the base models.

form by 2 points, the hallucination rate still remains 16.5%, indicating the difficulty of this task.

Table 5 shows example outputs of the base models. The input chat has a lot of distracting social chatter. The models filtered out social chatter as desired, but the outputs omit important information too. This problem is pronounced for GPT-2, which suffers from hallucination. More example outputs are in Table 9.

We discuss two main types of errors that should be addressed in future work for improved models.

**Error Type 1. Confusion over Deliberation:** We observe that BART-base tends to make errors when there is deliberation or discussion on slot values of the same type. Example 1 below contains a deliberation about departure dates ("Wednesday" vs. "Thursday"), and the incorrect one is picked up by the model (compare *True RW* (true rewrite) and *Pred RW* (predicted rewrite)). Similarly, in Example 2, both "Asian" and "Chinese" refer to food types, and the model takes the incorrect one.

### Example 1

| **User1**: | Can you help us get a train? We want to be leaving on Thursday. |
| --- | --- |
| **User2**: | No, actually, we need to go on Wednesday, I'll tell you why later. |
| **User1**: | Oh, okay then, Wednesday please. |
| *True RW*: | Can you help me find a train? I'll be traveling on Wednesday. |
| *Pred RW*: | Can you help me get a train? I want to be leaving on Thursday. |

### Example 2

| | Training | Testing |
|---|---|---|
| Flatten | \<user\>I would like to find a French restaurant, please.\<system\>You can try cote in the centre. Need a reservation?\<user\>If it's moderately priced, yes please. | \<user\>Shall we take a bus from Saint John's college to Pizza Hut Fen Ditton? No, too many stops. Let's take a taxi. Can you book a taxi for us? |
| Rewrite | \<user\>I would like to find a French restaurant, please.\<system\>You can try cote in the centre. Need a reservation?\<user\>If it's moderately priced, yes please. | \<user\>Book a taxi from Saint John's college to Pizza Hut Fen Ditton |
| Multi | \<user\>Hello! We're thinking of finding some European food. Did you want something in particular?\<user\>French food, if you can find it, please!\<system\>You can try cote in the centre. Need a reservation?\<user\>Ohh, that just opened I think, I hope its not too expensive.\<user\>We can ask, probably. Do you know if it is moderately priced? | \<user\>Shall we take a bus from Saint John's college to Pizza Hut Fen Ditton?\<user\>No, too many stops. Let's take a taxi.\<user\>Can you book a taxi for us? |

Table 6: Input formats for dialogue state tracking.

| | Intent | | | Inform | | | Request | | |
|---|---|---|---|---|---|---|---|---|---|
| | P | R | F1 | P | R | F1 | P | R | F1 |
| Flatten | 73.7 | 69.9 | 71.8 | 66.7 | **41.8** | **51.4** | 10.9 | 45.8 | 17.6 |
| Rewrite | **81.6** | **78.2** | **79.8** | 66.7 | 41.6 | 51.2 | **14.8** | **62.9** | **24.0** |
| Multi | 69.0 | 64.7 | 66.8 | **67.7** | 40.2 | 50.4 | 7.5 | 31.3 | 12.1 |

(a) In-domain accuracy.

| | Intent | | | Inform | | | Request | | |
|---|---|---|---|---|---|---|---|---|---|
| | P | R | F1 | P | R | F1 | P | R | F1 |
| Flatten | 53.1 | 53.1 | 53.1 | 18.8 | 11.7 | 13.3 | – | – | – |
| Rewrite | **62.3** | **62.3** | **62.3** | **25.0** | **19.0** | **20.6** | – | – | – |

(b) Unseen domain accuracy.

Table 7: Dialogue state tracking accuracy.

| | |
|---|---|
| **User1**: | It doesnt matter to me. |
| **User2**: | I think I feel like some Asian food. |
| **User1**: | Can you find some Chinese food? |
| *True RW*: | I'd like some Chinese food, please! |
| *Pred RW*: | It doesn't matter. I would like some Asian food. |

**Error Type 2. Early Memory:** We find that the model is less robust to informed slot values in latter utterances. The average number of turns per multi-user chat is 2.6. The average position of utterances, from which slot values are correctly recalled, is 1.99. However, the average utterance position for missed slot values is 2.6. This indicates that the model carries over slot values from earlier utterances better than from latter ones. Furthermore, when input contains dialogue history preceding the current turn, the average utterance positions of successfully recalled and missed slot values are 5.45 and 6.72, respectively, out of 6.54 utterances on average. The following example illustrates this error. The last user utterance clearly informs "moderate", yet the model ignores it.

**Example 3**

| | |
|---|---|
| **User1**: | I definitely want a certain range. |
| **User2**: | What do you think it should be? |
| **User1**: | Lets go for moderate. |
| *True RW*: | Yes definitely. I would like something moderate. |
| *Pred RW*: | I definitely want a certain range. |

### 5.3 Dialogue State Tracking

Dialogue state tracking is key to understanding user intents and relevant slot values. Assuming that dialogue state tracking occurs after each multi-user chat, we ran another experiment to verify that model-predicted rewrites of multi-user chats can benefit dialogue state tracking when dialogue systems are trained for *single-user* dialogues only. To simulate dialogue systems, we trained BART to take a user utterance and dialogue history as input and predict intents, informed slot values, and requested slot names separately (i.e., three separate models). The output is comma-separated values, e.g., "(train-day:wednesday),(train-departure:cambridge)" (Hosseini-Asl et al., 2020).

We explored three settings: **Flatten** and **Rewrite** simulate traditional dialogue systems and thus are trained on single-user dialogues using rewrites (not multi-user chats) in our data. When testing on multi-user dialogues, however, Flatten takes a flattened multi-user chat as if it is a "single" utterance from one user, whereas Rewrite takes BART-predicted rewrites (from the previous experiment) in place of the actual multi-user chats. **Multi** simulates a special dialogue system that is both trained and tested on multi-user dialogues. The input format for each setting is shown in Table 6.

Table 7a shows the precision, recall, and F1-

score of intents, informed slot values, and requested slot names. For the systems trained on single-user dialogues (Flatten and Rewrite), feeding predicted rewrites as input (Rewrite) substantially outperforms feeding a flattened multi-user chat (Flatten) in predicting intents (+8 points) and requests (+6.4 points); they perform similarly for inform prediction. This demonstrates the benefit of multi-user contextual query rewriting. While handling multi-user dialogues by summarizing them is an intuitive idea, no prior work has verified its effectiveness empirically and no public dialogue systems do this to our knowledge. Our work offers a dataset that enables this verification and verified its feasibility.

Interestingly, training a dialogue system directly on multi-user dialogues (Multi) underperforms Flatten by 5 points. This suggests that a simple seq2seq model and a medium-sized dataset are not enough to learn the dynamics of multi-user chats during training and that more research is needed to improve modeling. We believe our dataset can pave the way toward this research direction.

**Domain Transfer:** Here we verify that our dataset is also useful for handling multi-user dialogues in unseen domains. For this evaluation, we use the 62 multi-user chats collected in our pilot study (§3.1) as two unseen domains: finding gifts and finding news. As before, BART is trained for dialogue state tracking on our proprietary single-user dialogues for these two domains. During testing, the Flatten setting concatenates all utterances in the input multi-user chat as a "single" utterance. The Rewrite setting takes a rewrite predicted by the query rewriting model. It is important that this rewriting model is trained only on *our dataset* without exposure to any data from the unseen domains.

Table 7b shows accuracy on intent prediction and informed slot value prediction of the two settings. Accuracy on requested slots is not reported, since dialogues in these domains do not have requested slots. According to the overall scores, user dialogue state tracking in the unseen domains is generally more challenging than the eight domains in MultiWOZ. Nevertheless, Rewrite still outperforms Flatten for Intent by 9 points and for Inform by 7 points. This suggests that our dataset can assist dialogue systems to handle multi-user dialogues in even unseen domains via a query rewriting step in the language understanding pipeline.

# 6   Conclusion

We release the Multi-User MultiWOZ dataset, containing task-oriented dialogues between two users and one agent. The dialogues reflect important characteristics of collaborative decision-making in task-oriented settings, such as slot elicitation, social chatter, and deliberations. This dataset also enables the task of multi-user contextual query rewriting, which aims to rewrite a multi-user chat as a concise query that contains task-relevant information. We demonstrated that this task improves dialogue state tracking for a dialogue system trained for single-user dialogues both in-domain and cross-domain. This result is promising because this task can easily be plugged into a language understanding pipeline, processing multi-user dialogues with little modification to the dialogue system. Further, this task can be conducted on device, filtering out task-irrelevant information in multi-user chats and sending only task-relevant information to the system server for further processing, which enhances user privacy.

Our work assumes that a dialogue system starts dialogue state tracking only after a multi-user chat ends. This approach is common in practical dialogue systems, where systems start processing user inputs only when the user signals their utterance is directed at the device (e.g., by using a wake word). For multi-user task-oriented dialogues, an alternative approach is to track dialogue states after each utterance in multi-user chats; that is, a dialogue state is updated as soon as a user finishes speaking to the other user or the system. While this approach is theoretically plausible and allows dialogue systems to proactively intervene in the middle of a multi-user chat, it requires substantial effort to annotate a dialogue state for each user turn in multi-user chats. By contrast, query rewrites in our dataset are a byproduct of our data collection and thus do not require annotation effort. Nevertheless, it is an interesting question which method is more effective between turn-level dialogue state tracking and contextual query rewriting (as in our work). We leave this to future work.

## Limitations

We used a subset of dialogues in MultiWOZ 2.2 and the first four user utterances in each dialogue. Although this does not lose much information in terms of the diversity of services and slots, including more utterances for each dialogue would help train dialogue systems to be more robust to longer

conversations.

While the dialogues in our dataset reflect interesting social dynamics, some of the more challenging interactions, such as deliberation, are less frequent than other interactions. We can adjust the data collection manual to add more of such interactions. We leave this to future work.

## Ethics Statement

We tackle the problem of voice assistants processing a chat between users, and this could raise privacy concerns among the users. To alleviate this concern, we also proposed the task of multi-user contextual query rewriting, which is supported by our data. This allows user chats to be rewritten on device and only rewrites that contain task-relevant information to be sent to the server.

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

# A  Data Details

## A.1  Example Dialogues

### Example 1

| | |
|---:|:---|
| **User1**: | We need to book our train from cambridge. |
| **User2**: | Yes, we'd like a train that leaves after 14:30. |
| **Rewrite**: | I need to take a train from cambridge. I would like to leave after 14:30. |
| **System**: | I would be happy to help you with your request, first I will need to know your destination and what day you would like to travel. |
| **User2**: | Should we go on Tuesday or Wednesday? |
| **User1**: | Let's go on Wednesday. |
| **User2**: | Okay. Our destination is Peterborough. |
| **Rewrite**: | I've like to travel to Peterborough on Wednesday, please. |
| **System**: | I have on departing at 14:34. Would you like me to book it? |
| **User2**: | Yes, that sounds good. |
| **User1**: | Go ahead and book that train. |
| **User2**: | There are eight people in our party. |
| **Rewrite**: | Yes, for eight please. |
| **System**: | Your booking was successful. That is 132 GBP at the station. Is there anything else I can do for you today? |
| **User1**: | We should also find an expensive restaurant. |
| **User2**: | Please see if there are any options that serve eastern european food. |
| **Rewrite**: | Can you find me a place to eat? I would like an expensive eastern european restaurant. Thanks. |
| **System**: | I'm not showing any eastern european restaurants. Would you like to try a more specific type of food? |

### Example 2

| | |
|---:|:---|
| **User2**: | Hello. |
| **User1**: | Let me talk. |
| **User2**: | No, I've got this. |
| **User1**: | No. Can you direct us to the nearest police station? |
| **Rewrite**: | Can you direct me to the closest police station? |
| **System**: | Yes I can. The Parkside Police Station is located at Parkside, Cambridge. |
| **User1**: | Can you give us the phone number? |
| **User2**: | And the postcode. |
| **Rewrite**: | Can you please give me the postcode and phone number? |
| **System**: | The phone number is 01223358966 and the postcode is CB11JG. |
| **User1**: | Okay. |
| **User2**: | We got it, thanks. |
| **Rewrite**: | Ok, I've got it. Thanks. |
| **System**: | You are welcome. Have a nice day. |

### Example 3

| | |
|---:|:---|
| **User1**: | Hi Xavier, how are you? |
| **User2**: | Great, you ready to get this trip settled? You have the info? |
| **User1**: | I am and I do. We need an expensive hotel with free parking. |
| **Rewrite**: | I am looking for a hotel that is expensive and has free parking. |
| **System**: | I have about 5 great options for you. Do you prefer a certain area in the city? |
| **User2**: | I'll be on the east side that day. Can you do that? |
| **User1**: | Well, I'll be on the west but sure I can make it over there. |
| **User2**: | Great, and we'd like a guesthouse. |
| **Rewrite**: | I'm looking for a guesthouse in the east side of town. |
| **System**: | I am afraid I have nothing available with those specifications. Would you like a different are or a hotel? |
| **User1**: | Well damn. What should we do? |
| **User2**: | If not expensive then what about moderately priced you think? |
| **User1**: | That could work. Can you find that? |
| **Rewrite**: | Are there any moderately priced guesthouses in that part of town? |
| **System**: | Yes, both the Carolina B&B and the Warkworth House are moderately priced guesthouses on the east side. Would you like a room at one of these? |
| **User1**: | Great! I like the Warkworth House. |
| **User2**: | Well I am partial to the Carolina B&B. Let's see which have availability. |
| **User1**: | That's the best way to decide. |
| **User2**: | Do either have rooms for 5 peeps 5 nights beginning Tuesday? |
| **Rewrite**: | Yes, could you see if either of them have availability starting on Tuesday for 5 nights for 5 people? |
| **System**: | You have a reservation at the carolina bed and breakfast for Tuesday. Your reference number is BOHPJIFE |

## A.2 Distributions of Services, Intents, and Slots

We compare MultiWOZ 2.2 and MultiUserWOZ (our data) in terms of the distributions of services (Figure 2), intents (Figure 3), informed slots (Figure 4), and requested slots (Figure 5). They are counted at the turn level; active intents and informed/requested slots are counted for each turn, and a service is counted if it has any active intents for each turn.

The least common services in both datasets are bus, hospital, and police. These services are represented substantially more in MultiUserWOZ than MultiWOZ for both intents and informed/requested slots. We think this is desirable because it increases the exposure of these services to model training. Besides them, the attraction and restaurant services also have higher representation in MultiUserWOZ than MultiWOZ. The taxi, train, and hotel services have lower representation in MultiUserWOZ than MultiWOZ, but this does not seem problematic because their proportions are high.

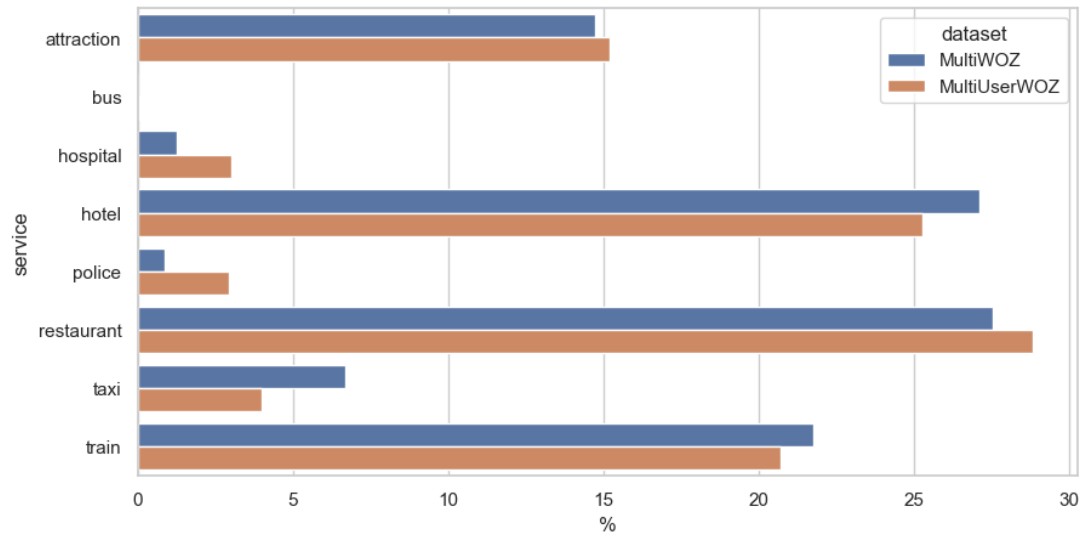

Figure 2: Distributions of services.

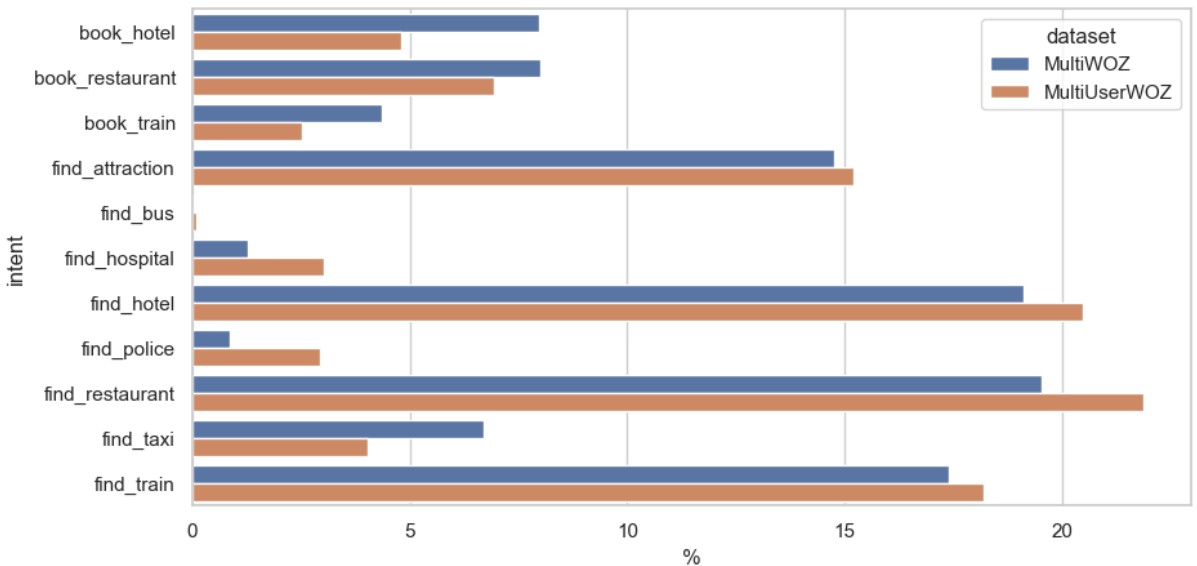

Figure 3: Distributions of intents.

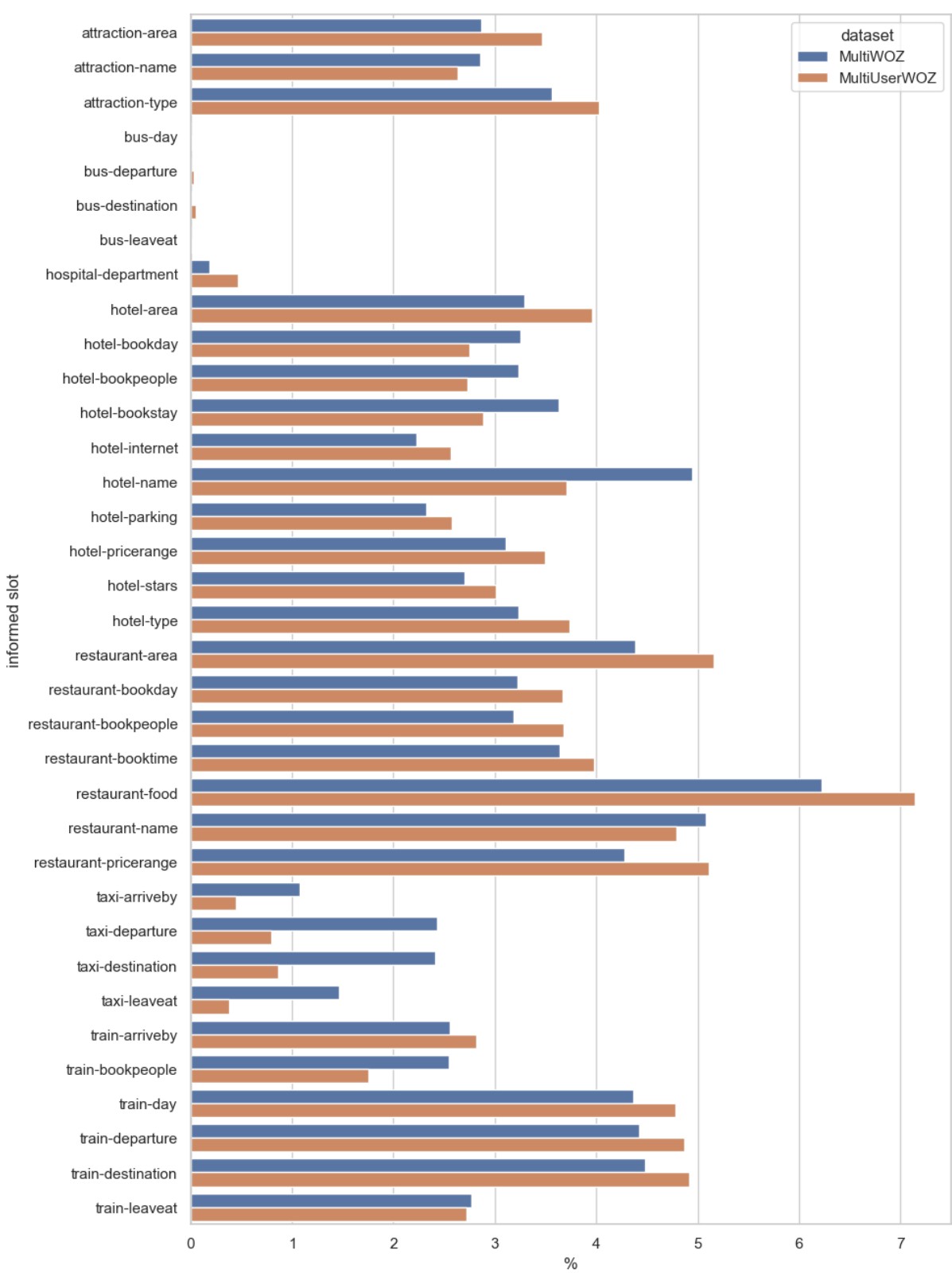

Figure 4: Distributions of informed slots.

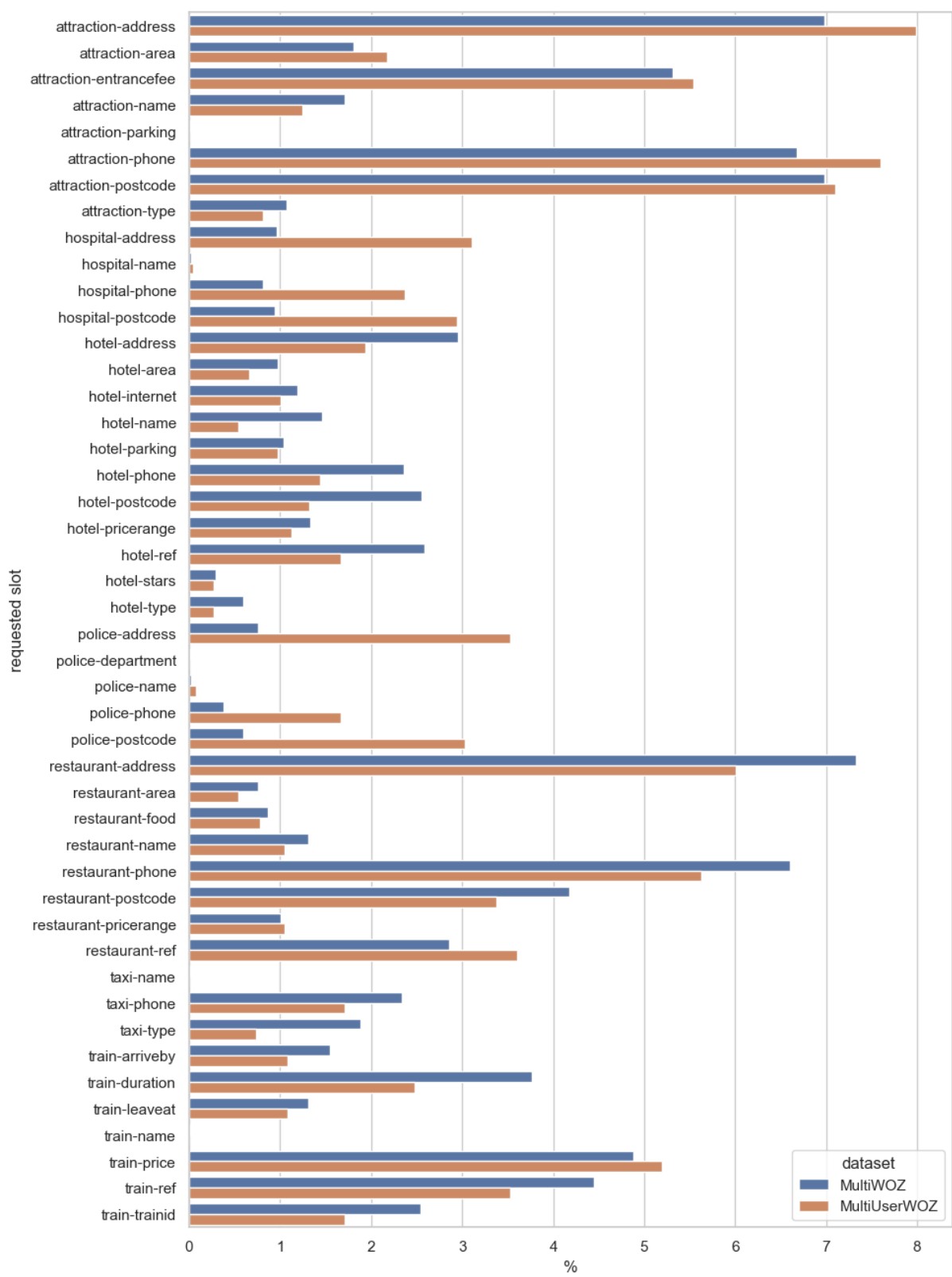

Figure 5: Distributions of requested slots.

### A.3 Dynamics Analysis

We analyzed three types of social dynamics in multi-user chats, namely, slot elicitation, social chatter, and deliberation.

Slot elicitation refers to users asking each other's preferences related to intents and slots. For example,
- "Let me think, there are how many in our group?"
- "Do you need any amenities?"
- "Do you think we should call first?"

Social chatter refers to user utterances that are not included in original source utterances but used for social conversation, such as jokes, feelings, etc. For example,
- "I'm not sure, but I've heard good things."
- "You don't have to tell the whole world how much money you have."
- "That's way too early for us."

Deliberation refers to making a decision over multiple options. For example,
- User1: "Either 4–5 not sure." → User2: "Better make it five to be safe."
- User1: "That might not be that bad." → User2: "Depends on the price" → User1: "It can't be that much."
- User1: "Do you think we should leave in the evening?" → User2: "I would prefer night time honestly."

To analyze the frequency of these dynamics in our data, we randomly sampled 100 multi-user chats for each split (a total of 300 chats). Next, a co-author of this paper marked each chat as to whether it contains slot elicitation, social chatter, and deliberation. Slot elicitation appears in 24%, social chatter 23%, and deliberation 2%.

## B  Data Collection Details

### B.1  Turker Qualifications and Compensation

We recruited turkers who satisfied the following criteria.

- Master Workers
- Number of HITs Approved $\geq 95$
- HIT Approval Rate (%) for all Requesters' HITs $\geq 95$

We paid \$2.00 for each HIT. Assuming each HIT takes 3 minutes, we believe this pay is reasonable.

### B.2  Tasks

Figure 6–8 show MTurk task pages for data collection.

Our instructions do not specify how the collected data would be used. But since we asked turkers to create stories, there is no risk of unintentional privacy breaches.

You will be given a conversation between a customer and an agent. The goal of this task is to convert this conversation to a conversation between **TWO** customers and the agent where the customers are **making decisions together**. Specifically, expand each utterance of the customer to a chat between two customers without changing the original intent of the utterance.

### Example

| | |
|---|---|
| **Original Utterance** | I need train reservations from `norwich` to `cambridge`. I'd like to know the `price` as well. |
| **Note** | Make sure to include preferences for ['train-arriveby'] if the original utterance contains them. |

↓↓↓ You should write customer utterances ↓↓↓

**Customer1**   We want to reserve a train. Are we leaving from norwich?
**Customer2**   Yes and get off at cambridge.
**Customer1**   Please let us know the price as well.

↑↑↑↑↑↑

**Agent**   I have 133 trains matching your request. Is there a specific day and time you would like to travel? They are all $50.

- Blue texts indicate preferences of the customers and red texts indicate information requested from the agent. These texts must be included in the expanded chat in their **literal form**.
- If the original utterance has a note, make sure to include the information requested in the note as long as the original utterance contains it. If the original utterance does not contain the requested information, you can ignore it.
- If the original utterance contains an answer to the agent's question (e.g. "Yes", "No"), make sure that the expanded chat too contains it.
- The last utterance of the expanded chat is meant to be spoken to the **agent**.
- Assume that the agent is listening to the chat between the customers. So, you don't have to repeat or summarize the chat when speaking to the agent.
- The expanded chat should naturally follow the past conversation and naturally lead to the agent's response.
- The agent sometimes does not provide some information requested in the original utterance if they can't. Do not omit a request in the original utterance just because the agent's response does not contain an answer.
- You may add some social chatter and deliberation to make the conversation natural (see the good example below). But the resulting chat should reflect the same preferences and requests as the original utterance -- no more, no less.

Figure 6: Data collection task instructions (1/2).

## Good Example

| | |
|---|---|
| **Original Utterance** | Yes, one more thing. We need a taxi that will take us from Clare Hall to our guesthouse. |

**Customer2**  Sounds good. Ah one more thing. Should we take a bus to get back to our guesthouse from Clare Hall?
**Customer1**  It's too much of a hassle. Let's take a taxi.
**Customer1**  Okay. Taxi please.

**Agent**  What time would you like that taxi?

- The original utterance contains an answer to the agent (i.e. "Yes") and the expanded chat also conveys it (i.e. "Sounds good").
- There is a deliberation (bus vs. taxi) and social chatter ("too much of a hassle"), but they do not change the original preferences and requests.

## Bad Example

| | |
|---|---|
| **Original Utterance** | No thanks. Can you book a train that goes to St. Louis Station . Can you tell me what time it arrives ? |

**Customer1**  What was the station we'll be going to? Oh St. Louis.
**Customer2**  I think we need two tickets. When will the train arrive?

**Agent**  Booking was successful, the total fee is 13.28 GBP payable at the station . Reference number is : LMZE8AMH .

- The original utterance begins with an answer to the agent (i.e. "No thanks") but this is missing in the expanded chat.
- The highlighted texts are rephrased ("St. Louis Station" to "St. Louis" and "arrives" to "arrive"), which should not.
- New information "two tickets" is introduced, which can be inconsistent with the agent's original behavior and response.

Figure 7: Data collection task instructions (2/2).

**Original Utterance** Hello, I would like information on the `Sesame Restaurant and Bar` .

**Note**

**Customer1** Write an utterance

**Customer2** Write an utterance

**Agent** Sesame Restaurant and bar number 01223358899 and it's address is 17 Hills Road City Centre.

---

**Original Utterance** I would like to make an appointment for `7` people at `6PM` on `Monday` .

**Note**

**Customer1** Write an utterance

**Customer2** Write an utterance

**Customer1** Write an utterance

**Customer2** Write an utterance

**Agent** I have booked your request for 7 people at the Sesame Restaurant and Bar, located at 17 Hills Road City Centre for 6PM on Monday.

---

**Original Utterance** Can I have the `reference number` please?

**Note** Make sure to include preferences for ['restaurant-booktime'] if the original utterance contains it/them.

**Customer1** Write an utterance

**Customer2** Write an utterance

**Agent** Yes certainly. However, there seems to be an error. I was unable to book that time. Is there another time I could book for you?

---

**Original Utterance** Is `5PM` available instead?

**Note**

**Customer1** Write an utterance

**Customer2** Write an utterance

**Customer1** Write an utterance

**Agent** Yes it is. I have booked a table for you at 17:00 on Monday for 7 people. The reference number is CZ3JGGJP .

Submit

Figure 8: Data collection task.

### B.3 Number of Generated Utterances and Speaker Labels

The number of generated utterances and speaker labels for each source utterance were predefined as follows. We stochastically sampled the number of generated utterances $k \in \{2, 3, 4\}$ where the weight for $k$ is $\frac{1}{|k-n|+1}$ and $n$ is the total number of informed or requested slots (so that $k$ is centered around this number). Further, we randomly chose Customer1 or Customer2 as the first speaker of the generated chat.

### B.4 Slot Constraints

When we displayed source user utterances to turkers, we highlighted informed slot values and requested slot names. We constrained that the highlighted texts be mentioned in generated multi-user chats. However, the slot values or names annotated in MultiWOZ do not always exactly match their expressions in user utterances. For example, informing hotel-parking:yes may be expressed as "I want free parking" rather than "yes", and train-bookpeople:1 may be expressed as "one ticket" rather than "1 ticket". Similarly, requesting train-ref may be expressed as "Give me a reference number", and restaurant-food may be expressed as "What kind of food do they serve?". Therefore, to capture informed slot values and requested slot names as much as possible, we added regex patterns as in Table 8.

When informed slot values or requested slot names cannot be captured, it is usually because they are not mentioned and in rare cases they are expressed in peculiar forms. We still provided turkers with the slot names and asked them to include information about those slots if found in the source utterance.

| (Slot Name, Value) | Added Patterns |
|---|---|
| (".*", "centre") | "center", "same area", "area" |
| (".*", "east") | "same area", "area" |
| (".*", "south") | "same area", "area" |
| (".*", "north") | "same area", "area" |
| (".*", "west") | "same area", "area" |
| (".*", "1") | "one", "same group of people" |
| (".*", "2") | "two", "same group of people" |
| (".*", "3") | "three", "same group of people" |
| (".*", "4") | "four", "same group of people" |
| (".*", "5") | "five", "same group of people" |
| (".*", "6") | "six", "same group of people" |
| (".*", "7") | "seven", "same group of people" |
| (".*", "8") | "eight", "same group of people" |
| (".*", "9") | "nine", "same group of people" |
| ("bookpeople", "1") | "for (me\|myself)", "me" |
| (".*", "guesthouse") | "guest house" |
| (".*", "swimmingpool") | "swimming pool", "pools", "pool" |
| (".*", "monday") | "same day" |
| (".*", "tuesday") | "same day" |
| (".*", "wednesday") | "same day" |
| (".*", "thursday") | "same day" |
| (".*", "friday") | "same day" |
| (".*", "saturday") | "same day" |
| (".*", "sunday") | "same day" |
| (".*", "expensive") | "same price range", "price range", "high priced", "high-end", "high end", "luxurious", "pricey", "up-scale", "a lot of money" |
| (".*", "cheap") | "same price range", "price range", "reasonable", "inexpensive", "af-fordable" |
| (".*", "moderate") | "same price range", "middle price range", "moderately priced", "price range", "isn't too high priced", "mid range", "not too expensive", "not cheap" |
| ("^hotel-internet$", "yes") | "free internet", "free wifi", "inter-net", "wifi", "wi-fi" |
| ("^hotel-parking$", "yes") | "free parking", "parking" |
| (".*", "dontcare") | "no (other \|particular )?prefer-ences?", "no particular", "(don't\|do not) (really \|actually )?have (a preference\|preferences)", "(don't\|dont\|do not) mind", "(doesn't\|does not\|don't\|do not) (re-ally \|actually )?(care\|matter)", "what ever", "whatever", "whenver", "not picky", "(don't\|do not\|doesn't\|does not) need" |

(a) Added patterns for informed slot values.

| Slot Name | Added Patterns |
|---|---|
| "-phone" | "(phone\|contact\|telephone) num-ber", "phone", "number" |
| "-postcode" | "postcodes?", "(post\|postal) codes?" |
| "-address" | "address(es)?", "postcode" |
| "-price" | "price range", "price" |
| "-ref" | "(ref\|reference\|confirmation) (number\|#)", "reference" |
| "-entrancefee" | "entrance fees?" |
| "train-duration" | "travel times?" |
| "taxi-type" | "car type" |
| "train-arriveby" | "arrival time", "arrives?", "when" |
| "train-leaveat" | "departure time", "depart", "leave", "when" |
| "train-trainid" | "train id", "trainid", "id" |
| "-area" | "what area", "exact area", "area" |
| "restaurant-food" | "food type", "(what )?(kind\|sort\|type) of food" |
| "attraction-type" | "(what )?types? of attraction", "attraction types?", "what types?", "types?" |
| "hotel-internet" | "(free )?(wi fi\|wi-fi\|wifi\|internet)" |
| "hotel-parking" | "(free )?parking" |
| "hotel-type" | "hotel type", "what type of ho-tels?" |
| "hotel-stars" | "how many stars", "stars of the hotel", "stars" |

(b) Added patterns for requested slot names.

Table 8: Added patterns for informed slot values and requested slot names.

## C   Data Validation Details

### C.1   Turker Qualifications and Compensation

For the validation task (§3.3), we first ran a qualification task, and Figure 9–12 show MTurk task pages for validator qualification. We recruited turkers using the same criteria as above. 13/55 qualified turkers participated in the main validation tasks.

We paid $0.50 for each HIT. Assuming each HIT takes 1 minute, we believe this pay is reasonable.

### C.2   Tasks

Figure 13–15 show MTurk task pages for validation.

Our instructions do not specify how the collected data would be used. But since we asked turkers to create stories, there is no risk of unintentional privacy breaches.

**Customer Chat**

**Customer1**   Let's visit a college in town.
**Customer2**   Sure, we will be around the west side of the town.

**Summary**   I want to find a college in the east.

Q1. Is the relationship between Customer1 and Customer2 really two customers making a decision together?

○ **Yes:** They are two customers making a decision together
○ **No:** Their relationship is something else, e.g., one is a customer and the other is an agent

Q2. Is the customer chat a realistic chat between humans?

○ **Very much:** Very realistic as a chat between humans
○ **Acceptable:** Acceptable as a chat between humans
○ **Definitely not:** Definitely not acceptable as a chat between humans

Q3. Is the last utterance of the customer chat realistic as an utterance toward the agent?

○ **Very much:** Very realistic as an utterance toward the agent
○ **Acceptable:** Acceptable as an utterance toward the agent
○ **Definitely not:** Definitely not acceptable as an utterance toward the agent

Q4-1. Is any important information in the customer chat missing in the summary to the extent that the agent may perform an action or provide information that is not intended by the customers? (See the instructions regarding what is acceptable and what is not)

○ **Yes:** Some important information in the chat is missing in the summary
○ **No:** No missing information

Q4-2. If you chose "Yes", specify what information is missing.

Q5-1. Does the summary contain additional information that is not present in the customer chat to the extent that the agent may perform an action or provide information that is not intended by the customers? (See the instructions regarding what is acceptable and what is not)

○ **Yes:** The summary contains additional information that is not present in the chat
○ **No:** The summary contains no additional information

Q5-2. If you chose "Yes", specify what additional information is present.

Figure 9: Validation qualification task (1/4). Turkers should choose "No" for Q4-1 or Q5-1 to pass.

| | |
|---|---|
| **Agent** | There are 5 colleges in the west. churchill college, clare college, clare hall, magdalene college, and queens' college. |

| | |
|---|---|
| **Customer Chat** | |
| **Customer2** | What is the entrance fee for Clare Hall? |
| **Customer1** | There is no entrance fee. |
| **Customer2** | What time does it open to the public? |
| **Customer3** | It's open from 9 to 6 to the public. Do you need any other information? |

| | |
|---|---|
| **Summary** | Could you tell me the entrance fee for clare hall please? |

Q1. Is the relationship between Customer1 and Customer2 really two customers making a decision together?

○ **Yes:** They are two customers making a decision together
○ **No:** Their relationship is something else, e.g., one is a customer and the other is an agent

Q2. Is the customer chat a realistic chat between humans?

○ **Very much:** Very realistic as a chat between humans
○ **Acceptable:** Acceptable as a chat between humans
○ **Definitely not:** Definitely not acceptable as a chat between humans

Q3. Is the last utterance of the customer chat realistic as an utterance toward the agent?

○ **Very much:** Very realistic as an utterance toward the agent
○ **Acceptable:** Acceptable as an utterance toward the agent
○ **Definitely not:** Definitely not acceptable as an utterance toward the agent

Q4-1. Is any important information in the customer chat missing in the summary to the extent that the agent may perform an action or provide information that is not intended by the customers? (See the instructions regarding what is acceptable and what is not)

○ **Yes:** Some important information in the chat is missing in the summary
○ **No:** No missing information

Q4-2. If you chose "Yes", specify what information is missing.

|  |
|---|

Q5-1. Does the summary contain additional information that is not present in the customer chat to the extent that the agent may perform an action or provide information that is not intended by the customers? (See the instructions regarding what is acceptable and what is not)

○ **Yes:** The summary contains additional information that is not present in the chat
○ **No:** The summary contains no additional information

Q5-2. If you chose "Yes", specify what additional information is present.

|  |
|---|

Figure 10: Validation qualification task (2/4). Turkers should choose "No" for Q1 to pass.

**Agent**      The entrance to Clare Hall is free. Anything else you need?

**Customer Chat**

**Customer1**   What is the address and phone number for Clare Hall?
**Customer2**   Please give me address and phone number for Clare Hall.
**Customer1**   Please provide address and phone number for clare hall.
**Customer2**   Phone number and addresss for Clare Hall, please.

**Summary**     Could you please provide me with the address and phone number?

Q1. Is the relationship between Customer1 and Customer2 really two customers making a decision together?

○ **Yes:** They are two customers making a decision together
○ **No:** Their relationship is something else, e.g., one is a customer and the other is an agent

Q2. Is the customer chat a realistic chat between humans?

○ **Very much:** Very realistic as a chat between humans
○ **Acceptable:** Acceptable as a chat between humans
○ **Definitely not:** Definitely not acceptable as a chat between humans

Q3. Is the last utterance of the customer chat realistic as an utterance toward the agent?

○ **Very much:** Very realistic as an utterance toward the agent
○ **Acceptable:** Acceptable as an utterance toward the agent
○ **Definitely not:** Definitely not acceptable as an utterance toward the agent

Q4-1. Is any important information in the customer chat missing in the summary to the extent that the agent may perform an action or provide information that is not intended by the customers? (See the instructions regarding what is acceptable and what is not)

○ **Yes:** Some important information in the chat is missing in the summary
○ **No:** No missing information

Q4-2. If you chose "Yes", specify what information is missing.

[                                                                                    ]

Q5-1. Does the summary contain additional information that is not present in the customer chat to the extent that the agent may perform an action or provide information that is not intended by the customers? (See the instructions regarding what is acceptable and what is not)

○ **Yes:** The summary contains additional information that is not present in the chat
○ **No:** The summary contains no additional information

Q5-2. If you chose "Yes", specify what additional information is present.

[                                                                                    ]

Figure 11: Validation qualification task (3/4). Turkers should choose "Definitely not" for Q2 to pass.

**Agent**      the phone number is 01223332360 and address isherschel road

**Customer Chat**

**Customer2**    What is a moderately priced restaurant in the area of Clare Hall?
**Customer1**    What do you want to eat?

**Summary**    What kind of moderately priced restaurants are in that area? I want to eat after I visit the college.

Q1. Is the relationship between Customer1 and Customer2 really two customers making a decision together?

○ **Yes:** They are two customers making a decision together
○ **No:** Their relationship is something else, e.g., one is a customer and the other is an agent

Q2. Is the customer chat a realistic chat between humans?

○ **Very much:** Very realistic as a chat between humans
○ **Acceptable:** Acceptable as a chat between humans
○ **Definitely not:** Definitely not acceptable as a chat between humans

Q3. Is the last utterance of the customer chat realistic as an utterance toward the agent?

○ **Very much:** Very realistic as an utterance toward the agent
○ **Acceptable:** Acceptable as an utterance toward the agent
○ **Definitely not:** Definitely not acceptable as an utterance toward the agent

Q4-1. Is any important information in the customer chat missing in the summary to the extent that the agent may perform an action or provide information that is not intended by the customers? (See the instructions regarding what is acceptable and what is not)

○ **Yes:** Some important information in the chat is missing in the summary
○ **No:** No missing information

Q4-2. If you chose "Yes", specify what information is missing.

[                                                                                    ]

Q5-1. Does the summary contain additional information that is not present in the customer chat to the extent that the agent may perform an action or provide information that is not intended by the customers? (See the instructions regarding what is acceptable and what is not)

○ **Yes:** The summary contains additional information that is not present in the chat
○ **No:** The summary contains no additional information

Q5-2. If you chose "Yes", specify what additional information is present.

[                                                                                    ]

Submit

Figure 12: Validation qualification task (4/4). Turkers should choose "Definitely not" for Q3 to pass.

You will be given a conversation among two customers and one agent, where the customers are (supposedly) making decisions together and getting help from the agent.

Each chat between the customers is labeled with a summary. Your task is to evaluate the quality of the chat and the summary.

**Example**

| Agent | Benny's is a good restaurant in the south area. Would you like me to make a reservation for you? |
|---|---|

**Customer Chat**

| Customer2 | **I'll love that**. Does Alice join us? |
|---|---|
| Customer1 | No, she needs to attend a business meeting. |
| Customer2 | Ok. We'd like to make a reservation for **2** people at 12:45 on **Monday**. |
| Customer1 | No, it's dinner, not lunch. **17:45** please. Give us their **phone number** too. |

| Summary | **That would be great**. Please make a reservation for **2** people for **17:45** on **Monday**. Give me their **phone number**. |
|---|---|

Q1. Is the relationship between Customer1 and Customer2 really two customers making a decision together?

⦿ **Yes:** They are two customers making a decision together
◯ **No:** Their relationship is something else, e.g., one is a customer and the other is an agent

Q2. Is the customer chat a realistic chat between humans?

⦿ **Very much:** Very realistic as a chat between humans
◯ **Acceptable:** Acceptable as a chat between humans
◯ **Definitely not:** Definitely not acceptable as a chat between humans

Q3. Is the last utterance of the customer chat realistic as an utterance toward the agent?

⦿ **Very much:** Very realistic as an utterance toward the agent
◯ **Acceptable:** Acceptable as an utterance toward the agent
◯ **Definitely not:** Definitely not acceptable as an utterance toward the agent

Q4-1. Is any important information in the customer chat missing in the summary to the extent that the agent may perform an action or provide information that is not intended by the customers? (See below regarding what is acceptable and what is not)

◯ **Yes:** Some important information in the chat is missing in the summary
⦿ **No:** No missing information

Q4-2. If you chose "Yes", specify what information is missing.

| |
|---|

Q5-1. Does the summary contain additional information that is not present in the customer chat to the extent that the agent may perform an action or provide information that is not intended by the customers? (See below regarding what is acceptable and what is not)

◯ **Yes:** The summary contains additional information that is not present in the chat
⦿ **No:** The summary contains no additional information

Q5-2. If you chose "Yes", specify what additional information is present.

| |
|---|

Figure 13: Validation task instructions (1/2).

**What is a good summary?**

Summaries have to

- Contain all essential information for the agent to perform an action or provide information intended by the customers, such as preferences for a search (e.g., "17:45", "Monday"), requested information (e.g., "phone number"), answers to the preceding question of the agent (e.g., "I'll love that").

Summaries **do not** have to

- Contain unimportant details in the chat, such as social chatter (e.g., "attend a business meeting") and declined preferences (e.g., "12:45").
- Use first-person plural instead of first-person singular. For example, it is fine if the summary contains "give me" in place of "give us" in the chat.
- Use the exact wording of agreement/disagreement. For example, it is fine if the summary contains "That would be great" or "Yes" in place of "I'll love that" in the chat.

Figure 14: Validation task instructions (2/2).

| Agent | The University Arms is an expensive, 4 star hotel with free wifi. Comparatively, the Alexander Bed and Breakfast is a cheaply priced guesthouse, also 4 stars. |
|---|---|

**Customer Chat**

| Customer2 | The University Arms will do. |
|---|---|
| Customer1 | Can you book that for us? |
| Customer2 | Yes, there'll be 8 of us, staying 3 nights, starting wednesday. |
| Customer1 | And also please give us the reference number. |

| Summary | Please book me some rooms for The University Arms to accommodate 8 people for 3 nights starting on wednesday. Can you also provide me the reference number after you book? |
|---|---|

Q1. Is the relationship between Customer1 and Customer2 really two customers making a decision together?

○ **Yes:** They are two customers making a decision together
○ **No:** Their relationship is something else, e.g., one is a customer and the other is an agent

Q2. Is the customer chat a realistic chat between humans?

○ **Very much:** Very realistic as a chat between humans
○ **Acceptable:** Acceptable as a chat between humans
○ **Definitely not:** Definitely not acceptable as a chat between humans

Q3. Is the last utterance of the customer chat realistic as an utterance toward the agent?

○ **Very much:** Very realistic as an utterance toward the agent
○ **Acceptable:** Acceptable as an utterance toward the agent
○ **Definitely not:** Definitely not acceptable as an utterance toward the agent

Q4-1. Is any important information in the customer chat missing in the summary to the extent that the agent may perform an action or provide information that is not intended by the customers? (See the instructions regarding what is acceptable and what is not)

○ **Yes:** Some important information in the chat is missing in the summary
○ **No:** No missing information

Q4-2. If you chose "Yes", specify what information is missing.

Q5-1. Does the summary contain additional information that is not present in the customer chat to the extent that the agent may perform an action or provide information that is not intended by the customers? (See the instructions regarding what is acceptable and what is not)

○ **Yes:** The summary contains additional information that is not present in the chat
○ **No:** The summary contains no additional information

Q5-2. If you chose "Yes", specify what additional information is present.

Figure 15: Validation task.

# D Experiment Details

## D.1 Model Details

BART-base, BART-large, T5-base, and GPT-2-base have 139M, 406M, 223M, and 124M parameters, respectively. For all experiments, we set the hyperparameters as follows: batch size=8, epoch=10, learning rate=2e-5, weight decay=0.01, and beam size=5. The input length is set to be 64 for BART-base, BART-large, and T5-base, and 128 for GPT2. Each task takes about 40 to 60 minutes for training on Tesla V100.

All experiments and models are developed with Huggingface Library licensed under Apache 2.0. Our models are fine-tuned based on publicly released GPT-2 (MIT License), BART-base and BART-large (Apache 2.0 License), and T5-base (Apache 2.0 License)

## D.2 Query Rewriting Results

Table 9 illustrates example outputs of the summarization models.

---

**User2:** I think I want to arrive in the evening.
**User1:** Yeah, but not after 9:00.
**User2:** Exactly.
**User1:** So the arrival time of 20:54 should work.
**Ground-Truth Rewrite:** Yes, I think the 20:54 arrival time should work.

**T5:** i want to arrive in the evening at 20:54. **(Good)**
**BART:** i want to arrive in the evening after 9:00. **(Picked up the wrong slot value)**
**GPT2:** i would like to arrive at 20:54 on sunday. **(Hallucination)**

---

**User1:** We don't travel much, so this is special.
**User2:** Yes, we have enough saved.
**User1:** We also need a good experience.
**User2:** Yes, so we will like it to be expensive.
**Ground-Truth Rewrite:** I would prefer that be expensive.

**T5:** i don't travel much, so this is special. i also need a good experience. **(Picked up social chatter and missed a slot value)**
**BART:** i don't travel much, so this is special. **(Picked up social chatter and missed a slot value)**
**GPT2:** yes, and i'd like to eat in a moderate price range. **(Hallucination and missing slot value)**

---

**User1:** Oh, we are so going to have fun there. Make sure they know we have 5 total guests we will need to make sure we get the reference number and book a room for 4 nights starting Monday.
**User2:** Okay that is right Monday will be great for me and others as well.
**User1:** Yes so I'm in for Monday just make sure we get reference number to confirm.
**User2:** Okay we will need the reference number for the stay starting Monday for 5 people for 4 nights.THanks.
**Ground-Truth Rewrite:** Yes. Please book me a room for 4 nights starting Monday. We have 5 total guests. I also need a reference number please.

**T5:** i would like to book a room for 5 people for 4 nights starting monday. **(Missed a requested slot)**
**BART:** yes please book it for 5 people and 4 nights starting monday. **(Missed a requested slot)**
**GPT2:** thursday from 3 pm and 3 people for 4 nights starting at 19:45 **(Hallucination, wrong slot values, and missing requested slot)**

---

**User1:** Thank you very much for all that information.
**User2:** We also need to make a booking for the restaurant in the West.
**User1:** Please make sure that it serves Italian food.
**User2:** And also make sure that the restaurant should be in the moderate price range.
**Ground-Truth Rewrite:** I need a a restaurant. The restaurant should be in the west and should serve italian food. The restaurant should be in the moderate price range

**T5:** thank you very much for all that information. i also need to make a booking for the restaurant in the west. please make sure that it serves italian **(Missed a slot value)**
**BART:** thank you very much. i also need to make a booking for the restaurant in the west. it serves italian food. **(Missed a slot value)**
**GPT2:** thank you, i also need to make a booking for the restaurant in the west and should serve italian food. **(Missed a slot value)**

---

Table 9: Example outputs of different models.