# OpenReview forum: "Multi-User MultiWOZ: Task-Oriented Dialogues among Multiple Users"
_EMNLP/2023/Conference — EMNLP 2023 Findings_

### Official Review · Reviewer_CX5P · 2023-07-31

**Soundness:** 4

**Excitement:**

3: Ambivalent: It has merits (e.g., it reports state-of-the-art results, the idea is nice), but there are key weaknesses (e.g., it describes incremental work), and it can significantly benefit from another round of revision. However, I won't object to accepting it if my co-reviewers champion it.

**Missing References:**

"Non-autoregressive dialog state tracking." ICLR (2020).
"UniConv: A unified conversational neural architecture for multi-domain task-oriented dialogues." EMNLP (2020).

**Paper Topic And Main Contributions:**

The authors propose a new type of task-oriented dialogue for multiple users. In this task, the authors extend the MultiWOZ benchmark from single-user dialogue to multiple-user dialogues. To do that, the authors transform each of the original user utterances into a short conversation between 2 users.

For this task, the authors propose a new model for contextual query rewriting to rewrite multiple-user queries to a concise task-oriented query so that the newly transformed queries are compatible with existing task-oriented dialogue systems.

**Questions For The Authors:**

- In Figure 1 (bottom, 2nd dialogue), the system seems to respond even without being prompted by users. Is this expected? In the real world, how does a system know when to respond without being prompted? For instance, in this example in Figure 1, the system can also respond right after user 2 utterance “Let’s do 19:45, Steve can’t get there till after 19:00”
- From a human user perspective, it is probably more sensible to replace “you” in current dialogues with some special token “<system>” to prompt the system to respond. The current way will limit the way users can interact with each other (e.g. how can a human user address the other human party?)

**Reasons To Accept:**

- A unique and interesting extension of the current task-oriented dialogues to enable interactions with multiple users in a single dialogue. Additional components are interesting, including social chatter and deliberation, and make the dialogues more natural.
- The data collection process is comprehensive. The data collection is grounded by a pilot study to reveal some key features in multi-user dialogues. There is also a validation round and quality assessment study to validate the generated dialogues based on a set of questionnaires and qualities.
- Good result analysis, including an analysis of query writing error types based on qualitative samples.

**Reasons To Reject:**

- In the data generation process, each annotator/turker is tasked to expand all user utterances in each task. It is would be more natural to have two different humans interact with each other (that can reach the same dialogue goal as before). The current approach might create data bias (e.g. confirmation bias). However, this bias seems to be minimal as shown (though not explicit) by the data validation results in Section 3.3.
- The current annotation of slot-value pairs, and intents are available after each multi-user chat, rather than after each user utterance. How would dialogue state tracking be applied here? How do we know when a chat is finished, then a dialogue state is required, and a system is expected to respond?
- The approach of contextual query rewriting is interesting and might be strong but it is not scalable to other domains and requires additional annotation of original single-user utterances to train such a model. Practically, those annotations are not available in many domains.
- I would advocate an approach to learn a multi-user dialogue state tracking model which can be considered a form of contextual rewriting (into compact slots and values). Here you would not assume access to the annotation of the original single-user utterances (which is harder to construct than dialogue state annotations).

**Reproducibility:**

4: Could mostly reproduce the results, but there may be some variation because of sample variance or minor variations in their interpretation of the protocol or method.

**Reviewer Confidence:**

4: Quite sure. I tried to check the important points carefully. It's unlikely, though conceivable, that I missed something that should affect my ratings.

---

> ### Author Rebuttal · Authors · 2023-08-28
>
> Thank you for the valuable comments.
>
> **1. In the data generation process, each annotator/turker is tasked to expand all user utterances in each task. It is would be more natural to have two different humans interact with each other (that can reach the same dialogue goal as before). The current approach might create data bias (e.g. confirmation bias). However, this bias seems to be minimal as shown (though not explicit) by the data validation results in Section 3.3.**
>
> We tried pairwise dialogue generation in our pilot study (Section 3.1), where we randomly paried up participants and let them naturally interact with each other via a texting interface. Unfortunately, the resulting dialogues were pretty poor and less realistic than dialogues generated by single annotators (lines 360-368). We speculate that this is because laypeople are usually not good at creating a collaborative scenario and holding a relevant dialogue in interactive settings. The same was observed even among family members who shared common social grounds. Tasking one annotator to generate the entire dialogue has been conducted in prior research as well to ensure quality. For instance, (1) Young et al. (2022), “Fusing Task-oriented and Open-domain Dialogues in Conversational Agents”, (2) Byrne et al. (2019), “Taskmaster-1: Toward a realistic and diverse dialog dataset”, (3) Nakamura et al. (2022), “HybriDialogue: An Information-Seeking Dialogue Dataset Grounded on Tabular and Textual Data”.
>
> **2. The current annotation of slot-value pairs, and intents are available after each multi-user chat, rather than after each user utterance. How would dialogue state tracking be applied here? How do we know when a chat is finished, then a dialogue state is required, and a system is expected to respond?**
>
> The dialogue state is needed when a user utterance is device-directed and should be replied to by the system. This requires a device-directedness detector (or more generally an addressee detector), which is far from a solved problem but beyond the scope of this paper. In essence, all other turns between users are considered as a side-discussion and are not processed separately by the dialogue system. Once a device-directed (DD) utterance is detected, the relevant context from the side-discussion (i.e., multi-user chat) between the users is extracted and added to the dialogue state. For reliable DD utterance detection, dialogue systems can use a predefined protocol that indicates the user prompting the system, e.g., by saying “go ahead Alexa/Siri” or by looking at the device which is detected by the device camera. We did not include this in our dataset because this is a simple matter of protocol that varies by different dialogue systems.
>
> Detecting the DD utterances based solely on semantics is more difficult and ambiguous. But this ambiguity is neither a deficit nor a completely new challenge to our dataset. It is the nature of any turn-taking dialogues with two or more speakers involved. While we usually assume utterance boundaries are given in single-user dialogue state tracking, in reality, detecting utterance boundaries is nontrivial due to the user mumbling, making corrections in the middle, etc., and practical systems usually have to rely on non-semantic cues anyway, such as a long pause, as a signal for the end of an utterance.
>
> Tracking dialogue states after each multi-user chat has some limitations as well, e.g., does not allow for many types of proactivity in the middle of the user-user chat. But it is a good compromise to stimulate multi-party interaction research. Alternatively one could parse a multi-user chat and track dialogue states at the utterance level within the chat as the reviewer seems to suggest; e.g., the intents in a multi-user chat might have to be augmented, and part of the corpus should be manually reannotated. This approach would be an interesting direction and we discuss this more in detail in 4 below.
>
> **3. The approach of contextual query rewriting is interesting and might be strong but it is not scalable to other domains and requires additional annotation of original single-user utterances to train such a model. Practically, those annotations are not available in many domains.**
>
> While it is true that the accuracy of contextual query rewriting on unseen domains may not be as good as in-domain accuracy (as in most NLP tasks indeed), we verified its scalability in Section 5.3 (lines 528-554) by showing that unseen domains (GiftFinder and NewsFinder) that are substantially different from training domains still benefit from contextual query rewiring in dialogue state tracking. Furthermore, as the capacity of language models increases, it is very likely that their ability to rewrite queries would generalize better across domains.
>
> The query rewriting approach has additional advantages over other approaches. For instance, compared to training a dialogue state tracking model directly on multi-user chats, it performs better and helps preserve user privacy as emphasized in the paper. Dialogue state tracking within a multi-user chat, which is another alternative approach, requires even more annotation effort than query rewriting, partly because rewrite annotations in our dataset are a byproduct of our dialogue generation. That said, we are eager to see more research come up with better solutions, and we believe our dataset would serve as a valuable testbed. As an example research direction, co-authors have been able to generate multi-user dialogues for new domains automatically from single-user data, using a combination of rule-based expansion followed by paraphrasing.
>
> **4. I would advocate an approach to learn a multi-user dialogue state tracking model which can be considered a form of contextual rewriting (into compact slots and values). Here you would not assume access to the annotation of the original single-user utterances (which is harder to construct than dialogue state annotations).**
>
> We assume that the reviewer is proposing utterance-level dialogue state tracking within a multi-user chat. This approach makes sense since it can hold rich information, such as different points of view of the users (different values for the same slot per user), meta-intents (e.g., talking about the performance of the system, talking about turn-taking), etc. While we believe this approach would be fruitful and its effectiveness is worth testing, this annotation is a significant investment. First and foremost, the proposed approach needs multi-user task-oriented dialogues to start with. But these are already a very scarce resource in the first place. It is indeed one of our contributions to obtain such dialogues by expanding single-user dialogues (which are comparably abundant) and publicly release them. Moreover, query rewriting annotation is a byproduct of this process, so no additional effort is needed on top of this dialogue generation.
>
> Even if we assume there are other cheap sources of multi-user task-oriented dialogues, we would humbly disagree that the utterance-level annotation of intents and slots is easier than query rewriting. First, annotators have to annotate dialogue states after every turn, which induces substantial cognitive load. Second, annotation of dialogue states is prone to errors, as shown by the fact that the original MultiWOZ had so many annotation errors and the SGD dataset avoided dialogue state annotation completely by using dialogue simulation. Lastly, it increases the dialogue state tracking model’s dependency on the domain schema since it has to predict domain-specific intents and slots for each utterance. By contrast, the query rewrite annotation requires reading the entire chat and rewriting it as a summary only once per multi-user chat. That said, we think that it would be an interesting methodological question which method is more scalable, reliable, and effective in dialogue state tracking.
>
> **5. In Figure 1 (bottom, 2nd dialogue), the system seems to respond even without being prompted by users. Is this expected? In the real world, how does a system know when to respond without being prompted? For instance, in this example in Figure 1, the system can also respond right after user 2 utterance “Let’s do 19:45, Steve can’t get there till after 19:00”**
>
> Please refer to our answer to 1.
>
> **6. From a human user perspective, it is probably more sensible to replace “you” in current dialogues with some special token “<system>” to prompt the system to respond. The current way will limit the way users can interact with each other (e.g. how can a human user address the other human party?)**
>
> When users talk to voice assistants in general and in most task-oriented dialogue datasets, they call the systems “you” instead of a system token or the system name (e.g., “can YOU recommend books for kids?”). In our pilot study as well, participants naturally used "you" when speaking to the voice assistant. Hence, it is more reflective of user behavior to use “you” to refer to both other users and the system. This of course increases the complexity of the problem, but we think this is something we need to solve rather than avoid. For instance, we could introduce an explicit prompting protocol (e.g., a wake word), which is easy to add to our dataset, detect who the user looks at when they speak, and even improve semantic understanding.
>
> **7. Missing References:
> "Non-autoregressive dialog state tracking." ICLR (2020). "UniConv: A unified conversational neural architecture for multi-domain task-oriented dialogues." EMNLP (2020).**
>
> We will discuss these papers in the Related Work section in the final paper.

---

### Official Review · Reviewer_6huk · 2023-08-03

**Soundness:** 3

**Excitement:**

3: Ambivalent: It has merits (e.g., it reports state-of-the-art results, the idea is nice), but there are key weaknesses (e.g., it describes incremental work), and it can significantly benefit from another round of revision. However, I won't object to accepting it if my co-reviewers champion it.

**Paper Topic And Main Contributions:**

This paper releases the Multi-User MultiWOZ dataset, containing task-oriented dialogues between two users and one agent and proposes the multi-user contextual query rewriting task that rewrites multi-user chats as concise task-oriented requests.

**Reasons To Accept:**

The proposed Multi-User MultiWOZ dataset in this paper can aid researchers in conducting research in this field. The introduction of multiple users amplifies the technical challenges faced by the NLU and DST modules of dialogue systems. Furthermore, this was validated through experiments conducted in the study.

**Reasons To Reject:**

The constructed Multi-User MultiWOZ dataset has certain limitations. Firstly, it only involves changing one user to two users instead of incorporating more users. Secondly, multiple users do not interact separately with the dialogue system; instead, after the two users finish their conversation, the system interacts with them as a whole. This design may not pose sufficient challenges for the dialogue management module.

**Reproducibility:**

5: Could easily reproduce the results.

**Reviewer Confidence:**

4: Quite sure. I tried to check the important points carefully. It's unlikely, though conceivable, that I missed something that should affect my ratings.

---

> ### Author Rebuttal · Authors · 2023-08-28
>
> Thank you for the valuable comments.
>
> **1. Firstly, it only involves changing one user to two users instead of incorporating more users.**
>
> We agree that incorporating more than two users would be nice and would be fruitful work. We want to highlight, though, that multi-user task-oriented dialogue is a completely new problem and there is no benchmark dataset at all. While our dataset is limited to two users, we believe our work, the first of its kind, has made meaningful contributions: defining the problem, releasing a good-size dataset, introducing a related task of contextual query rewriting, and verifying its effectiveness in dialogue state tracking. Our dataset would serve as a valuable first step and benefit the research and industry communities for improved datasets.
>
> **2. Secondly, multiple users do not interact separately with the dialogue system; instead, after the two users finish their conversation, the system interacts with them as a whole. This design may not pose sufficient challenges for the dialogue management module.**
>
> The source utterance of each multi-user chat is exactly the case of each user interacting with the system separately. Therefore, we do not need to generate such utterances again. One can simply train a dialogue system on a mix of multi-user chats and source utterances to cover both user-user chats and user-agent 1:1 chats.

---

### Official Review · Reviewer_bSQy · 2023-08-06

**Soundness:** 3

**Excitement:**

3: Ambivalent: It has merits (e.g., it reports state-of-the-art results, the idea is nice), but there are key weaknesses (e.g., it describes incremental work), and it can significantly benefit from another round of revision. However, I won't object to accepting it if my co-reviewers champion it.

**Paper Topic And Main Contributions:**

This paper releases the Multi-User MultiWOZ dataset: task-oriented dialogues among two users and one agent and proposes the novel
task of multi-user contextual query rewriting.

**Reasons To Accept:**

1. This paper releases the Multi-User MultiWOZ dataset, task-oriented dialogues between two users and one agent.
2. This paper propose the multi-user contextual query rewriting task that rewrites multi-user chats as concise task-oriented requests.

**Reasons To Reject:**

1. In comparison to regular task-oriented dialogues, multi-user task-oriented dialogues have their own characteristics. Should a specific evaluation metric be designed to address these characteristics?

**Reproducibility:**

4: Could mostly reproduce the results, but there may be some variation because of sample variance or minor variations in their interpretation of the protocol or method.

**Reviewer Confidence:**

3: Pretty sure, but there's a chance I missed something. Although I have a good feel for this area in general, I did not carefully check the paper's details, e.g., the math, experimental design, or novelty.

---

> ### Author Rebuttal · Authors · 2023-08-28
>
> **1. In comparison to regular task-oriented dialogues, multi-user task-oriented dialogues have their own characteristics. Should a specific evaluation metric be designed to address these characteristics?**
>
> Thank you for bringing up this point. Yes, we agree that multi-user task-oriented dialogues have unique characteristics and need related evaluation metrics. We indeed introduced several criteria and metrics with which we evaluated and analyzed the generated dialogues in the paper and verified that our dataset has high quality. First, in Section 3.3, we used five criteria to measure (a) whether the users’ relationship is two customers making decision together, (b) whether the chat is a realistic chat between humans, (c) whether the last utterance in the chat is a realistic utterance toward the agent, (d) whether the rewrite has any missing information that is present in the chat, (e) whether the rewrite contains any information that is absent in the chat. All these criteria except (b) are specific to multi-user task-oriented dialogues, and our evaluation result shows the high quality of our dataset: (a) 0.99/1, (b-c) 1.8/2, (d-e) 0.98/1 (lines 282-288).
>
> In addition, in Section 3.4, we measured five important quantities related to multi-user task-oriented dialogues. (a) instances of slot elicitation, (b) instances of social chatter, (c) instances of deliberation, (d) number of entities, (e) number of negations. In our dataset, (a) and (b) appear in 24% and 23% of dialogues, which are reasonably high, while (c) is quite low at 2% (lines 297-298). (d) and (e) are 1.2-1.9x as high as the original MultiWOZ (Table 2), which reveals that our dataset is semantically richer and dialectic than single-user dialogues.
>
> We believe some of these metrics can be automated quite reliably. For instance, we can build a classifier that computes the probability that two users are customers (vs. having another relationship) using existing dialogue datasets, and classifiers that compute the probability that a dialogue contains slot elicitation, social chatter, or deliberation respectively. We can also use a public library, e.g., spaCy, to count named entities and negations, as we already did in Section 3.4. We will add this discussion in the final paper.

---

### Meta-Review · Area_Chair_jetL · 2023-10-06

**Recommendation:** 4

**Metareview:**

The dataset introduced in the paper is an interesting contribution to the field of dialogue systems. It addresses a need for realistic, multi-user task-oriented dialogues and offers a comprehensive data collection process and insightful result analysis. This dataset is likely to aid researchers in advancing the development of dialogue systems for complex, real-world scenarios.

---

### Decision · Program_Chairs · 2023-10-07

**Decision:**

Accept-Findings

**Comment:**

The dataset introduced in the paper is an interesting contribution to the field of dialogue systems. It addresses a need for realistic, multi-user task-oriented dialogues and offers a comprehensive data collection process and insightful result analysis. This dataset is likely to aid researchers in advancing the development of dialogue systems for complex, real-world scenarios.